# Depletion of Ric-8B leads to reduced mTORC2 activity

**Maíra H. Nagai**[1], **Victor P. S. Xavier**[1], **Luciana M. Gutiyama**[1¤], **Cleiton F. Machado**[1], **Alice H. Reis**[2], **Elisa R. Donnard**[3], **Pedro A. F. Galante**[3], **Jose G. Abreu**[2], **William T. Festuccia**[4], **Bettina Malnic**[1] *

**1** Department of Biochemistry, University of São Paulo, São Paulo, Brazil, **2** Institute of Biomedical Sciences, Federal University of Rio de Janeiro, Rio de Janeiro, Brazil, **3** Centro de Oncologia Molecular, Hospital Sírio-Libanês, São Paulo, Brazil, **4** Department of Physiology and Biophysics, Institute of Biomedical Sciences, University of São Paulo, São Paulo, Brazil

¤ Current address: Centro de Transplante de Medula Óssea, Instituto Nacional do Câncer, Rio de Janeiro, Brazil
* bmalnic@iq.usp.br

**Data Availability Statement:** All relevant data are within the manuscript and its Supporting Information files.

**Funding:** MHN received support from Fundação de Amparo à Pesquisa do Estado de São Paulo (#14/

## Abstract

mTOR, a serine/threonine protein kinase that is involved in a series of critical cellular processes, can be found in two functionally distinct complexes, mTORC1 and mTORC2. In contrast to mTORC1, little is known about the mechanisms that regulate mTORC2. Here we show that mTORC2 activity is reduced in mice with a hypomorphic mutation of the Ric-8B gene. Ric-8B is a highly conserved protein that acts as a non-canonical guanine nucleotide exchange factor (GEF) for heterotrimeric Gαs/olf type subunits. We found that Ric-8B hypomorph embryos are smaller than their wild type littermates, fail to close the neural tube in the cephalic region and die during mid-embryogenesis. Comparative transcriptome analysis revealed that signaling pathways involving GPCRs and G proteins are dysregulated in the Ric-8B mutant embryos. Interestingly, this analysis also revealed an unexpected impairment of the mTOR signaling pathway. Phosphorylation of Akt at Ser473 is downregulated in the Ric-8B mutant embryos, indicating a decreased activity of mTORC2. Knockdown of the endogenous Ric-8B gene in cultured cell lines leads to reduced phosphorylation levels of Akt (Ser473), further supporting the involvement of Ric-8B in mTORC2 activity. Our results reveal a crucial role for Ric-8B in development and provide novel insights into the signals that regulate mTORC2.

## Author summary

Gene inactivation in mice can be used to identify genes that are involved in important biological processes and that may contribute to disease. We used this approach to study the Ric-8B gene, which is highly conserved in mammals, including humans. We found that Ric-8B is essential for embryogenesis and for the proper development of the nervous system. Ric-8B mutant mouse embryos are smaller than their wild type littermates and show neural tube defects at the cranial region. This approach also allowed us to identify the biological pathways that potentially contribute to the observed phenotypes, and uncover a

15495-8, MN). BM and VPSX receive support from Fundação de Amparo à Pesquisa do Estado de São Paulo (#16/24471-0 (BM) and #2019/05166-0 (VPSX)). BM and LMG received support from Conselho Nacional de Desenvolvimento Científico e Tecnológico (CNPq). The funders had no role in study design, data collection and analysis, decision to publish, or preparation of the manuscript.

novel role for Ric-8B in the mTORC2 signaling pathway. mTORC2 plays particular important roles in the adult brain, and has been implicated in neurological disorders. Our mutant mice provide a model to study the complex molecular and cellular processes underlying the interplay between Ric-8B and mTORC2 in neuronal function.

## Introduction

Ric-8B (resistant to inhibitors of cholinesterase 8B) is a highly conserved protein which interacts with Gαs class subunits from heterotrimeric G proteins [1,2]. *In vitro*, Ric-8B can work as a guanine nucleotide exchange factor (GEF) for both Gαs and Gαolf [1,3]. While Gαs is ubiquitously expressed, Gαolf is restrictedly expressed in the olfactory neurons and in a few regions of the brain, such as the striatum [4–6]. Ric-8B expression in adult mice is highly predominant in the same tissues where Gαolf is expressed indicating that these two proteins are functional partners *in vivo* [2]. Consistent with the role of a GEF, Ric-8B is able to amplify odorant receptor signaling or dopamine receptor signaling through Gαolf in cultured cells [2,7–9]. Also, a series of studies indicate that Ric-8B regulates Gα protein abundance in the cells, and suggest that Ric-8B may serve as a chaperone that promotes Gα protein stability and the formation of functional G protein complexes [7,8,10–13].

In addition to the full-length Ric-8B, an alternatively spliced version of Ric-8B lacking exon 9, denominated Ric-8BΔ9, is also highly expressed in the olfactory epithelium. Differently from full-length Ric-8B, Ric-8BΔ9 does not bind to Gαs and does not show GEF activity, or does it very inefficiently [2,3]. Studies have shown that both Ric-8B and Ric-8BΔ9 are able to interact with the different Gγ subunit types, Gγ13, Gγ7 and Gγ8 [10]. Chan and colleagues showed that Ric-8BΔ9, but not full-length Ric-8B, can bind Gβ1γ2 [3]. These results suggest that besides acting on the Gαs subunits, the Ric-8B proteins may also play a role in Gβγ signaling.

Despite the restricted pattern of expression in adult mice, previous studies have shown that complete knockout of the Ric-8B gene results in mice that are not viable and that die early during embryogenesis (between E4 and E8.5) [7]. Here we investigated the physiological roles of Ric-8B during development using a gene trapped allele of Ric-8B that shows reduced levels of Ric-8B expression. We found that the Ric-8B mutant embryos are small, fail to close the neural tube at the cephalic region and die around E10.5. In the embryo, Ric-8B gene expression is predominant in the nervous system, more specifically in the neural folds in the cephalic region and in the ventral region of the neural tube. Increased apoptosis is observed in the region of the neural tube defects in the Ric-8B mutant embryos. Comparative transcriptome analysis unexpectedly revealed that mTOR signaling is impaired in the Ric-8B mutant embryos. mTOR is a serine/threonine protein kinase that acts as the catalytic core of two distinct complexes: mTORC1 and mTORC2. mTORC1 mainly controls cell growth and metabolism, promotes protein synthesis and is the best characterized complex to date [14,15]. mTORC2, on the other hand, has been implicated in the regulation of cytoskeletal organization, cell survival and cell migration [14–19]. Both complexes, mTORC1 and mTORC2, have been linked to the control of protein synthesis, although the role of mTORC2 is not as clearly defined as that of mTORC1 [20,21]. We found that mTORC2 activity, but not mTORC1 activity, is downregulated in the mutant embryo. Similar effects were observed in HEK293T and HepG2 cells which were knocked down for Ric-8B. Altogether these results show that Ric-8B is essential for embryogenesis. They also show that depletion of Ric-8B reduces mTORC2 activity.

## Results

### Generation of Ric-8B gene trap mice

In order to generate mice that are deficient for Ric-8B we obtained two Baygenomics ES cell lines [22], which contain a gene trap vector in the Ric-8B gene. In the RRH188 cell line, the vector is inserted in the intron between exons 3 and 4, and in the RRA103 cell line the vector is inserted in the intron between exons 7 and 8 (Fig 1A). We used these two ES cell lines to produce chimeric mice, but we could only obtain chimeras from the RRH188 cell line. The insertion of the gene trap vector leads to the expression of a chimeric mRNA containing exons 1, 2 and 3 of the Ric-8B gene in frame with the β-geo sequence [23]. The resulting Ric-8B fusion protein is likely to be nonfunctional, because it only contains amino acids 1–246 from Ric-8B. Chimeric males were crossed with C57BL/6 females and the agouti-colored offspring were analyzed for transmission of the gene trap vector. As expected, approximately 50% of these mice were heterozygous for the gene trap insertion. These mice develop normally with no signs of deficits, when compared to their wild type siblings.

### Ric-8B mutant mice are embryonic lethal

Heterozygous mice were intercrossed to generate homozygous mutant mice. Genotyping of the offspring revealed the absence of mice homozygous for the Ric-8B gene mutation (Table 1). In order to determine the time of embryonic death, embryos from heterozygous intercrosses were genotyped at different developmental stages (Fig 1B, Table 1). We found that homozygous embryos die around embryonic day 10.5.

As mentioned above, two major transcript forms of Ric-8B are expressed in the olfactory epithelium, a full-length Ric-8B isoform and Ric-8BΔ9, an isoform which lacks exon 9 (Fig 1A) [2]. RT-PCR experiments showed that the Ric-8B gene is expressed in the mouse embryo at early stages of development, and that Ric-8BΔ9 is the predominant isoform present in the embryo, while both isoforms are equally abundant in the olfactory epithelium of adult mice (Fig 1C, [2]).

RT-PCR analysis demonstrated that even though detectable, the levels of Ric-8B mRNA are much lower in the homozygotes (Ric-8B[bgeo/bgeo]) than in the wild type (Ric-8B[wt/wt]) or heterozygous (Ric-8B[wt/bgeo]) embryos (Fig 1D and 1E). Accordingly, Western blot analysis of extracts prepared from whole-embryos demonstrated that the levels of Ric-8B protein are lower in the homozygous embryos than in wild type or heterozygous embryos (Fig 1F). The residual levels present in the homozygous embryos is likely to result from the gene knockout technology used, that is, a small fraction of the mRNA can be produced by splicing out the intron containing the gene trap vector leading to the production of wild type Ric-8B mRNAs [23].

### Ric-8B gene expression in the adult mouse

We used the β-geo gene reporter, which is expressed under the control of the Ric-8B promoter, to monitor the Ric-8B gene expression in the heterozygous (Ric-8B[wt/bgeo]) mice. Strong blue staining is detected in the olfactory epithelium, but not in the olfactory bulb, vomeronasal organ or brain (Fig 2A). Staining can also be detected in the septal organ of Masera, an isolated small patch of sensory epithelium located at the ventral base of the nasal septum [24], which is known to contain olfactory sensory neurons that express Gαolf as well as other canonical olfactory sensory neuron genes [25]. Although no staining was observed in the medial view of the brain (Fig 2A), when the brain is sectioned in a parasagittal plane, blue staining of the striatum is revealed (Fig 2B). X-gal staining of sections cut through the nasal cavity shows that β-galactosidase activity is present throughout the olfactory epithelium and in the region where

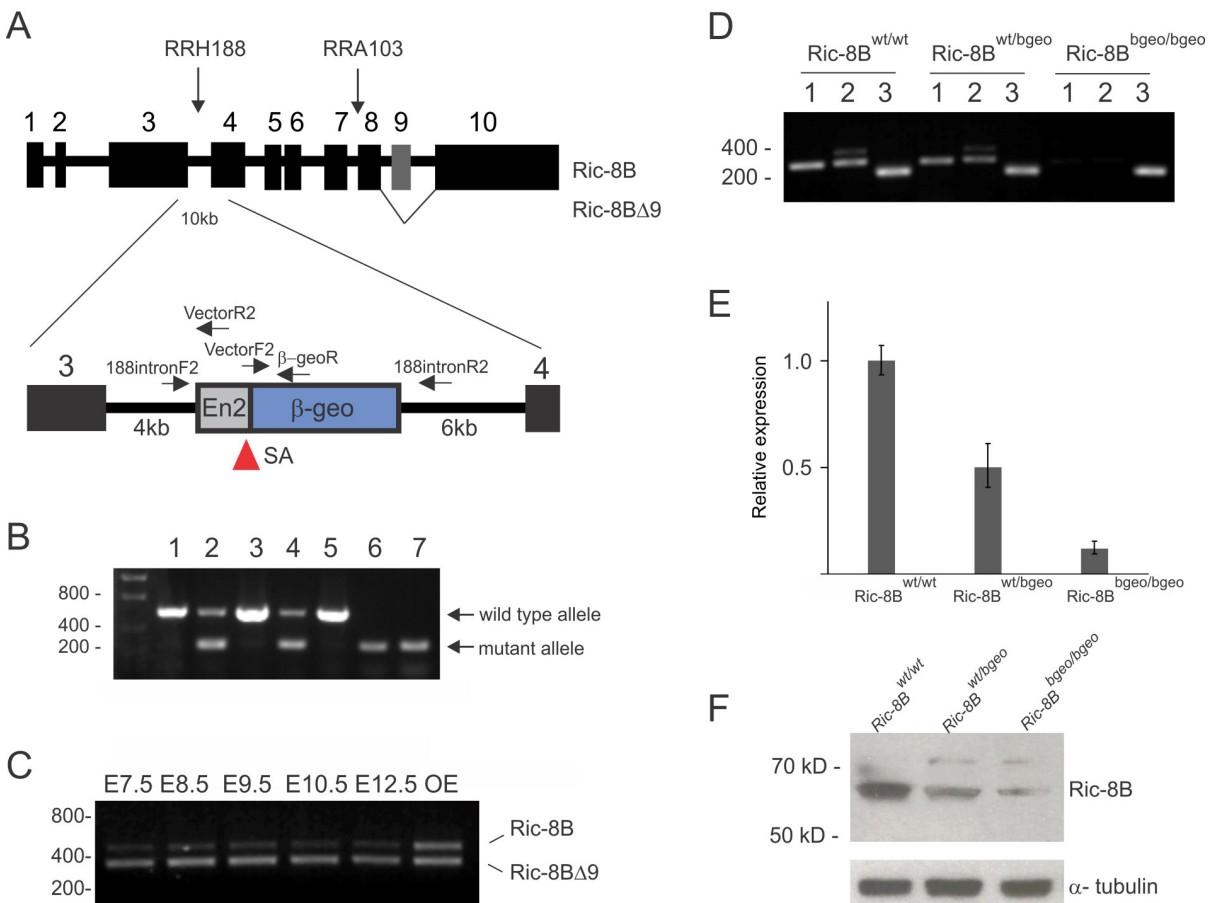

**Fig 1. Ric-8B gene trap mice.** (A) The genomic structure of the Ric-8B gene is shown, with its ten exons (1–10) and nine introns (between the exons) and the insertion sites of the gene trap vector in the ES cell lines RRH188 and RRA103. The insertion of the gene trap vector in intron 3 in the RRH188 cell line leads to the expression of chimeric mRNA containing exons 1, 2 and 3 in frame with the β-geo sequence. The locations of the primers used for PCR-based genotyping are indicated. SA: splice acceptor site. The Ric-8BΔ9 isoform lacks exon 9 (indicated in grey), is indicated. B) Multiplex PCR-based genotyping of the embryos obtained from intercrossing of heterozygous mice using one forward primer (188intronF2) and two reverse primers (VectorR2 and 188intronR2). A 312 bp PCR product is expected for the mutant allele, and a 582 bp PCR product is expected for the wild type allele. Representative analyzed embryos showed the following genotypes: Ric-8B$^{wt/wt}$ (wild type, lanes 1, 3 and 5), Ric-8B$^{wt/bgeo}$ (heterozygote, lanes 2 and 4) and Ric-8B$^{bgeo/bgeo}$ (homozygous, lanes 6 and 7). (C) Ric-8B gene expression in the mouse embryo. RT-PCR was conducted to amplify Ric-8B and Ric-8BΔ9 transcripts from RNA prepared from wild type mouse embryos at different developmental stages. The PCR product sizes expected using the pair of primers that flank the ninth exon (A) are 462 bp (Ric-8B) and 342 bp (Ric-8BΔ9). (D) RT-PCR was conducted to amplify the regions between exon 3 and exon 5 (1) and exon 7 and exon 10 (2) of Ric-8B and actin (3) from embryos with different genotypes as indicated. (E) Real-time PCR was conducted to compare the expression levels of the Ric-8B gene in wild type (Ric-8B$^{wt/wt}$), heterozygotes (Ric-8B$^{wt/bgeo}$) and homozygous (Ric-8B$^{bgeo/bgeo}$) mutant embryos. Transcript levels were normalized to β-actin levels and are shown relative to the expression levels in wild type embryos. (F) Western blot analysis of protein extracts from embryos with the different genotypes using anti-Ric-8B antibodies. α-tubulin was used as a loading control.

the neurons of the septal organ are located (Fig 2C, 2D and 2F). These results are in agreement with analysis of Ric-8B gene expression by *in situ* hybridization [2], and indicate that the expression of the β-geo reporter is indistinguishable from that of the endogenous Ric-8B gene.

We also examined the Ric-8B gene expression in different tissues of the heterozygous mice. No staining was observed in any of the several analyzed tissues (see S1 Fig). Altogether, these results confirm the previous findings that the expression of the Ric-8B gene is predominantly expressed in a few tissues in the adult mouse. Lower levels of Ric-8B gene expression are

**Table 1. Analysis of offspring and embryos from intercrosses of heterozygous mice.** Genotypes were determined by PCR. The number of abnormal or dead embryos is shown in parenthesis.

| Stage | Ric-8B$^{wt/wt}$ | Ric-8B$^{wt/bgeo}$ | Ric-8B$^{bgeo/bgeo}$ (abnormal) | Total # |
|---|---|---|---|---|
| E7.5 | 11 | 22 | 9 | 42 |
| E8.5 | 31 | 56 | 22(20) | 109 |
| E9.5 | 100 | 164 | 97(82) | 361 |
| E10.5 | 8 | 40 | 25(23) | 73 |
| E12.5 | 7 | 12 | 1(1) | 20 |
| Post natal | 29 | 53 | 0 | 82 |

however also detectable in several other tissues, as shown in publicly available gene expression data, such as the EMBL-EBI expression Atlas (https://www.ebi.ac.uk/gxa/home) and Mouse ENCODE transcriptome data (https://www.ncbi.nlm.nih.gov/gene/237422/).

## Ric-8B is required for embryonic growth and development of the cranial neural tube

Analysis of the heterozygous embryos showed that expression from the Ric-8B gene promoter is restricted to the cephalic neural folds and neural tube regions (Fig 3B, 3E and 3E'). Notably, the levels of β-galactosidase expression are significantly higher in the Ric-8B$^{bgeo/bgeo}$ embryos when compared to heterozygote embryos (Fig 3C, 3F and 3F'). These results are expected, since the Ric-8B$^{bgeo/bgeo}$ embryos carry two copies of the β-geo reporter gene while the Ric-8B$^{wt/bgeo}$ embryos have only one copy.

At E8.5–9.5, heterozygous embryos (Ric-8B$^{wt/bgeo}$, Fig 3B, 3E and 3E') are morphologically similar to that of the wild type embryos (Ric-8B$^{wt/wt}$, Fig 3A, 3D and 3D'). However, from E8.5, great part of homozygous mutant embryos (Ric-8B$^{bgeo/bgeo}$; Fig 3B and 3C) are slightly smaller and show phenotypic abnormalities in the prosencephalon. The reduced size of homozygote embryos is emphasized in later stages (Fig 3J), and the failure in the closure of the cephalic neural tube becomes evident at E9.5 (Fig 3F and 3F'). In order to better visualize the fusion of the cephalic neural folds we performed scanning electron microscopy. Wild type as well as heterozygous E9.5 embryos show normally closed neural tube (Fig 3G and 3G', 3H and 3H'). The Ric-8B$^{bgeo/bgeo}$ embryos, however, do not display fused midline hemispheres (Fig 3I and 3I'). These embryos are not able to close the neural tube and usually display open rhombencephalic, mesencephalic and prosencephalic vesicles (Fig 3I and 3I'). It is important to note that this 'open brain' phenotype is highly penetrant, shown by ~86% of the analyzed embryos.

Expression of the Ric-8B gene along the embryonic anterior-posterior axis was analyzed in transverse sections of X-gal stained embryos. Staining in Ric-8B$^{wt/bgeo}$ embryos is highly restricted to the notochord, dorsal neural tube in the region of the presumptive brain, and to the ventral region of the neural tube, including the floor plate of the spinal cord (Fig 4A, 4B and 4E–4H). The same regions are strongly stained in Ric-8B$^{bgeo/bgeo}$ embryos, however, less intense staining is also detected all over the neural tube and regions of the adjacent mesoderm (Fig 4C, 4D and 4I–4L). As mentioned above, while the neural tube is normally closed in the Ric-8B$^{wt/bgeo}$ embryos, it fails to close in the brain region in Ric-8B$^{bgeo/bgeo}$ embryos (arrows in Fig 4G and 4K).

Ric-8B expression pattern in the notochord (Fig 4B) and the floor plate (Fig 4H, S4A Fig) is highly reminiscent of the sonic hedgehog (*Shh*) expression at the same embryonic stages [26]. However, we found that Shh signaling is not grossly altered in the homozygous Ric-8B mutant embryos (S4B and S4C Fig). *In situ* hybridization on sections of an E10.5 wild type embryo shows that Gαolf is not co-expressed with Ric-8B in the floor plate, while Gαs is highly

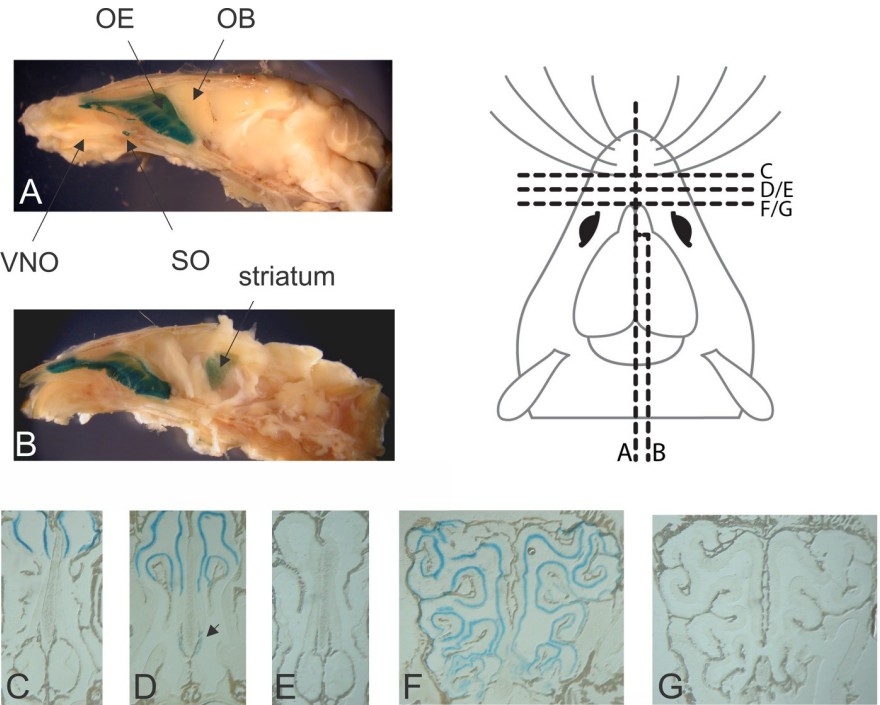

**Fig 2. Ric-8B expression in the adult Ric-8B^{wt/bgeo} mouse as revealed by X-gal staining.** (A) Sagittal whole-mount view of the nasal cavity and brain stained with X-gal. Blue staining can be detected in the olfactory epithelium (OE) and septal organ (SO), but not in the olfactory bulb (OB) nor in the vomeronasal organ (VNO). (B) The brain region is cut in a parasagittal plane, revealing blue staining of the striatum. (C-G) X-gal staining of sections cut through the nasal cavity of Ric-8B ^{wt/bgeo} mice (C, D and F) or Ric-8B ^{wt/wt} mice (E and G). X-gal staining is present throughout the olfactory epithelium (C, D and F), and in the septal organ region (arrow in D). In the same experimental conditions, no staining is observed in the wild type tissues. Sections in A-G were taken from the regions indicated in the schematic representation of the mouse brain.

expressed all over the neural tube (S5A Fig). These results suggest that Gαs may be the target for Ric-8B in the mouse embryo, instead of Gαolf.

The expression of the β-geo reporter gene in the neural folds and roof plate at E8.5-E9.5 (Fig 4) strongly suggest that the deficiency of Ric-8B gene expression is leading to the failure of neural tube closure. Previous studies have shown that this phenotype can result from a variety of embryonic disturbances [27,28], such as abnormalities in the contraction of apical actin microfilaments within neuroepithelial cells [27], or reduced/ increased apoptosis of neuroepithelial cells [27]. We analyzed the distribution of polymerized actin in the neural tubes of wild type and mutant E9.5 embryos, however, no significant differences between wild type and mutant embryos were observed (Fig 5A). Disturbances in apoptosis were analyzed by immunostaining for activated caspase-3 in E9.5 embryo sections. We found an increased number of apoptotic cells in the neural tube of Ric-8B^{bgeo/bgeo} embryos, as well as in the cranial mesenchyme (Fig 5B).

We also tested the impact of Ric-8B gene depletion on apoptosis *in vitro*, by using mouse embryonic fibroblasts (MEFs) generated from the Ric-8B mutant embryos. Even though Ric-8B is predominantly expressed in the nervous system (as shown in Figs 3 and 4), MEFs prepared from wild type embryos also express Ric-8B. Noticeably, MEF preparations from Ric-8B^{bgeo/bgeo} embryos died within few days in culture. We found that, while the number of dividing cells seems to be unaltered, as revealed by bromodeoxyuridine (BrdU) staining, the

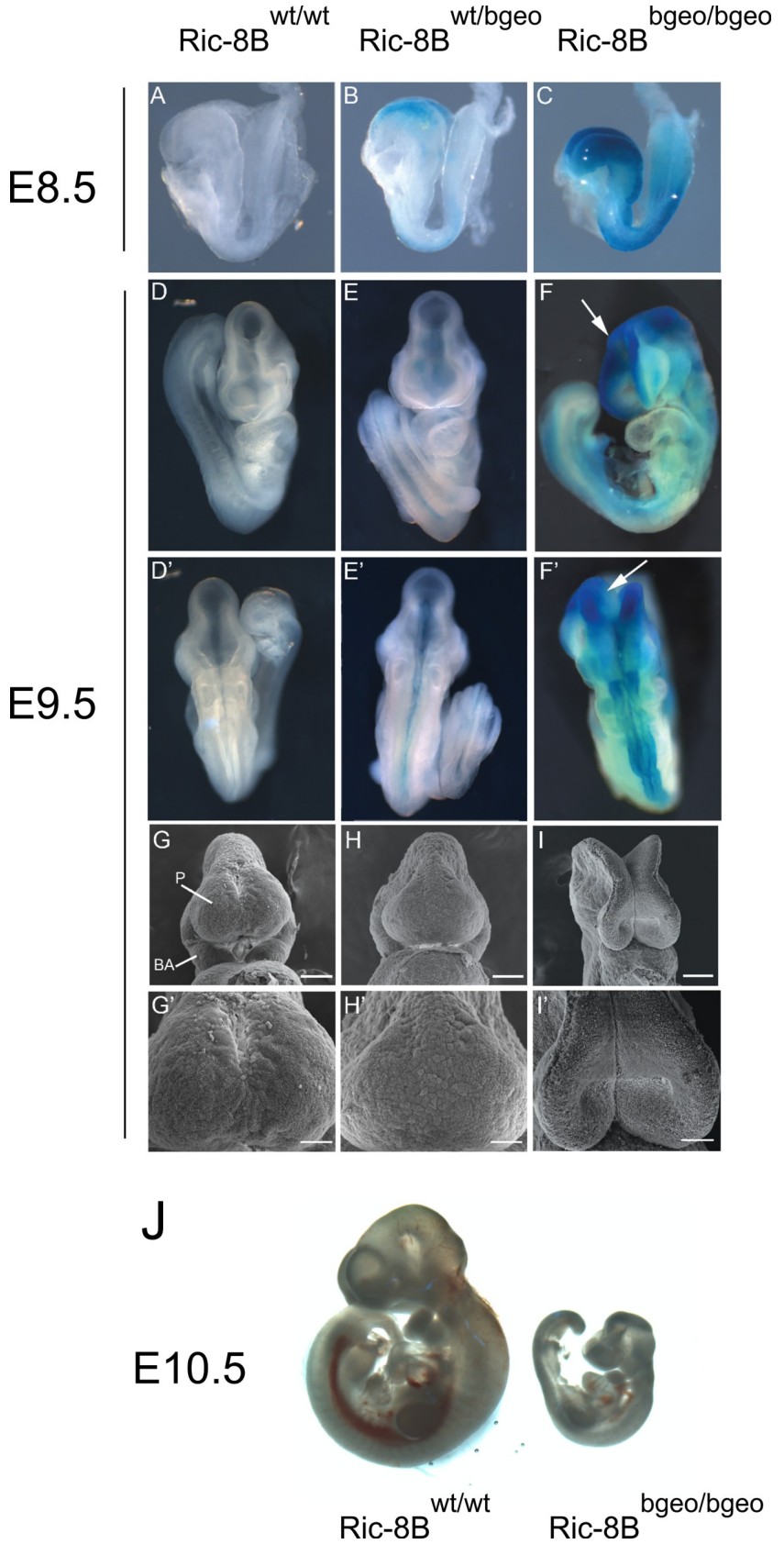

**Fig 3. Ric-8B mutant embryos are smaller and show defective neural tube closure in the cephalic region.** (A-F′) Wild type (Ric-8B^wt/wt), heterozygous (Ric-8B^wt/bgeo) or homozygous (Ric-8B^bgeo/bgeo) embryos at E8.5 or E9.5 stained with X-gal are shown (not shown in scale). Ventral (D-F) and dorsal (D'-F') views of the embryos are shown. In the Ric-8B^bgeo/bgeo embryos, the cephalic neural folds are not fused (white arrows) but the other regions of the neural tube are normally closed (F, F'). (G-I) Wild type (Ric-8B^wt/wt), heterozygous (Ric-8B^wt/bgeo) or homozygous (Ric-8B^bgeo/bgeo) embryos were examined by scanning electron microscopy at E9.5. G', H' and I' are magnified regions from G, H and I, respectively. P (prosencephalon), BA (first branchial arch). Scale bars in G- I represent 100 μm, in G'-I', 50 μm. (J) Representative images of a wild type and a homozygous mutant embryo (shown in scale).

number of cells immuno stained for activated caspase-3 is increased in MEFs generated from Ric-8B^bgeo/bgeo embryos, when compared to MEFs generated from Ric-8B^wt/wt embryos (Fig 5C). These results indicate that depletion of Ric-8B leads to increased apoptosis.

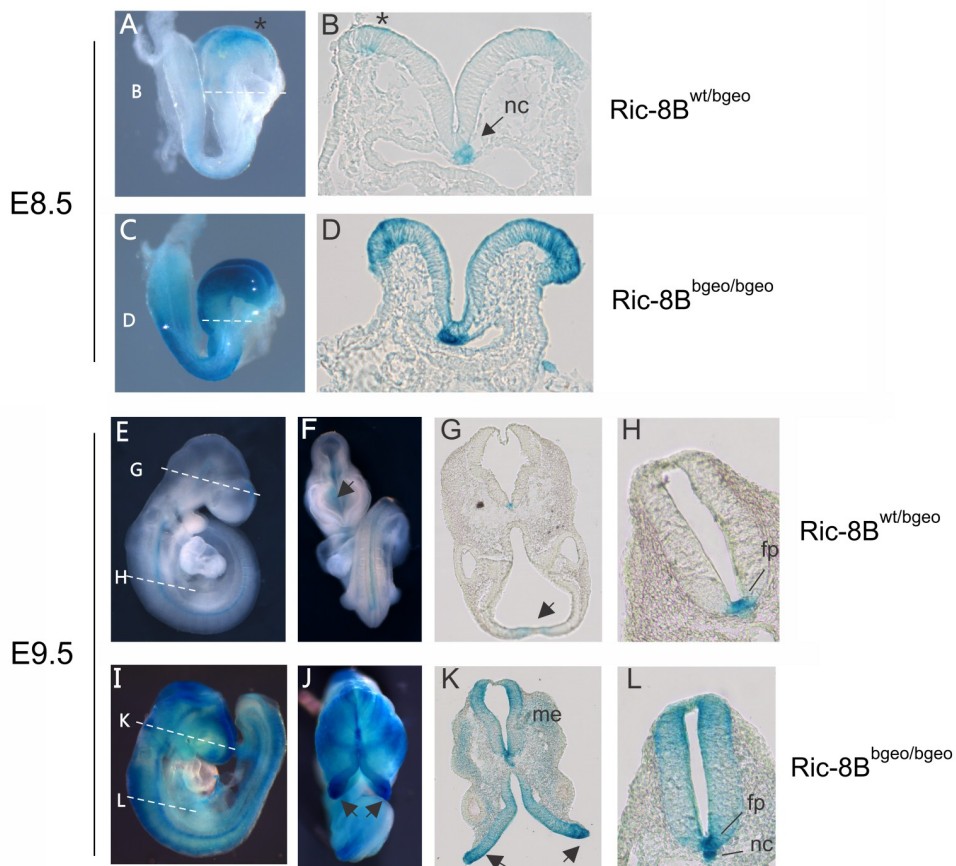

**Fig 4. Ric-8B expression in the embryo is predominant in the nervous system.** Upper panels: sagittal views of a Ric-8B^wt/bgeo embryo (A) or a Ric-8B^bgeo/bgeo embryo (C) at E8.5. The lines indicate the levels at which the sections were cut (shown in B and D). In the Ric-8B^wt/bgeo embryo (A and B), blue staining is detected in the neural folds in the brain region (asterisk) and in the notochord (nc). In the Ric-8B^bgeo/bgeo embryo (C and D) blue staining is detected in the same regions as in the Ric-8B^wt/bgeo embryo, however the staining is stronger and expanded. Lower panels: sagittal and ventral views of a Ric-8B^wt/bgeo embryo (E and F) or a Ric-8B^bgeo/bgeo embryo (I and J) at E9.5. In the Ric-8B^wt/bgeo embryo (E-H), blue staining is detected in the neural fold fusion in the brain region (arrows) and in the floor plate region (fp). Note that the neural tube is closed (arrow in G). (H) Higher magnification of the neural tube in the Ric-8B^wt/bgeo embryo showing strong blue staining in the floor plate region and weaker staining in the notochord. In the Ric-8B^bgeo/bgeo embryo (I- L) blue staining is detected in the same regions as in the Ric-8B^wt/bgeo embryo; however, the staining is stronger and expanded. Note that the neural tube is not closed in the anterior region of the head (arrows in K) but are normally closed at the more caudal regions. (L) Higher magnification of the neural tube in the Ric-8B^bgeo/bgeo embryo showing strong blue staining in both the floor plate (fp) and notochord (nc) regions. Me (mesenchyme).

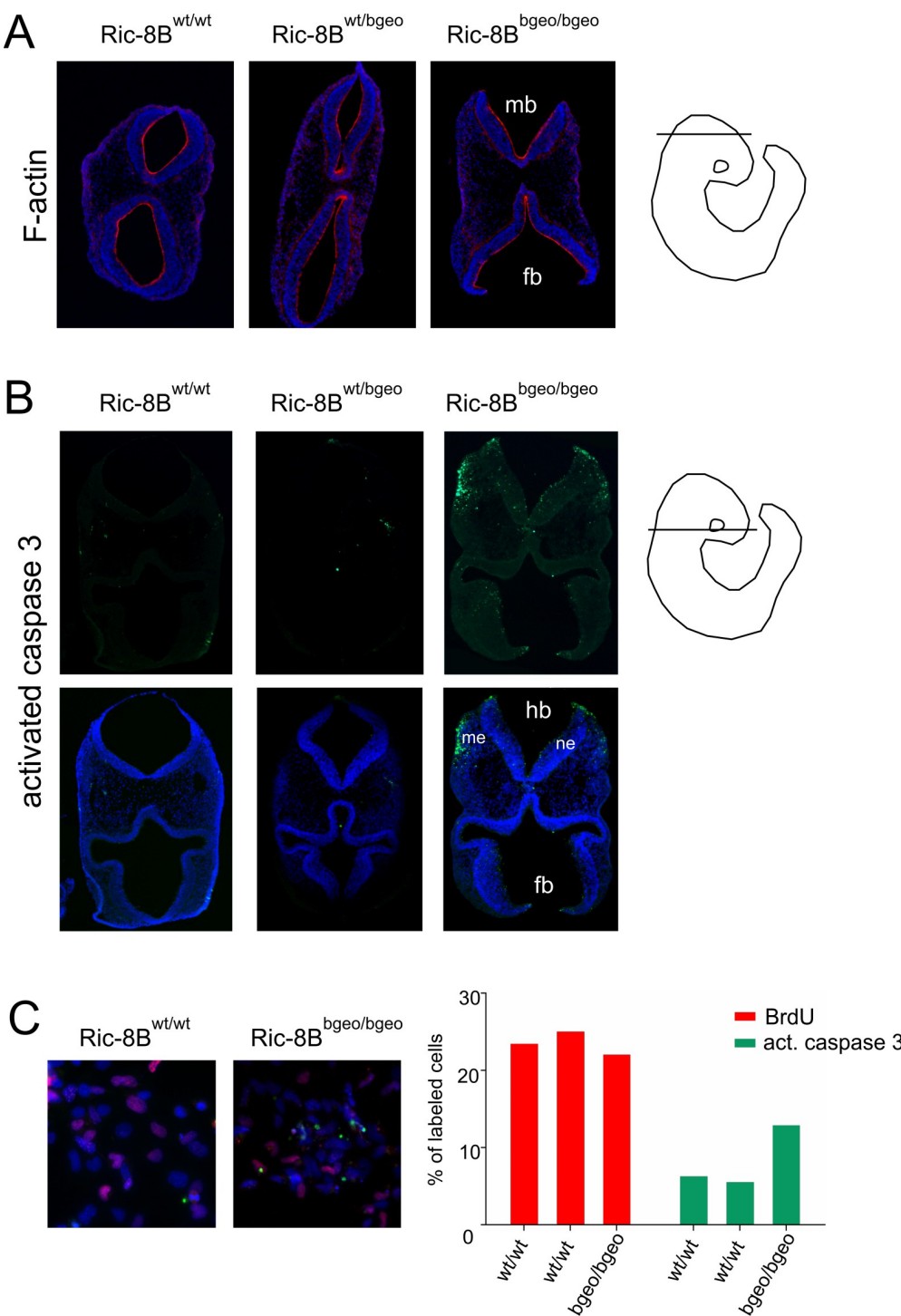

**Fig 5. Increased apoptosis in Ric-8B$^{bgeo/bgeo}$ embryos and MEFs.** (A) Transverse sections of E9.5 embryos stained with rhodamine-conjugated phalloidin (red) show that actin filament (F-actin) is highly concentrated in the apical region of the neural tube in wild type, heterozygous and mutant embryos. (B) Transverse sections of E9.5 embryos stained for active caspase 3 (green). Ric-8B$^{bgeo/bgeo}$ embryos show an increased number of apoptotic cells in the mesenchyme and neuroepithelium when compared to wild type or heterozygous embryos. DAPI was used to stain the nuclei. Forebrain (fb); mesenchyme (me); midbrain (mb); hindbrain (hb); neuroepithelium (ne). The approximate localizations of the sections are indicated to the right. (C) MEFs were generated from Ric-8B$^{wt/wt}$ (n = 2) or Ric-8B$^{bgeo/bgeo}$ (n = 1) embryos were double stained for BrdU (red) and activated caspase-3 (green). The percentages of cells labeled in each case are indicated in the graph.

## Cell signaling pathways altered in the Ric-8B mutant embryos

To gain insight into the molecular mechanisms impacted by the mutation in the Ric-8B gene, we sequenced and compared the transcriptomes of mutant and wild type E10.5 embryos. A total of 947 coding genes were found to be differentially expressed (FDR < 0.05) (S1 Table and S2 Table). The results of gene ontology analysis (Enrichr, http://amp.pharm.mssm.edu/Enrichr/, [29]) are consistent with the mutant embryo phenotype, showing enrichment of Mammalian Phenotype terms related to lethality and decreased size (S3 Table, adjusted p-value <0.05) and Biological Process terms related to nervous system development, apoptosis and mostly to mRNA catabolic process and translation (S4 Table, adjusted p-value <0.05).

To determine the most significantly affected pathways in the Ric-8B^bgeo/bgeo embryo, and therefore to be able to explore the molecular role of Ric-8B in this biological context, we used Ingenuity Pathway Analysis (IPA) (Fig 6A). Consistent with the role played by Ric-8B in G protein function, multiple pathways related to GPCRs and G protein signaling such as G Beta Gamma signaling, Gα12/13 signaling, α-Adrenergic signaling and Phospholipase C signaling were identified (S5B Fig).

Since Ric-8B was shown to regulate protein stability of Gαs in heterologous cells, we analyzed Gαs expression in the mutant embryos. As shown in Fig 6B, while extracts from Ric-8B^wt/wt and Ric-8B^wt/bgeo embryos show one singular band at the expected MW size for Gαs, the extracts from Ric-8B^bgeo/bgeo embryos show an additional band with a higher MW. These results suggest that the Gαs protein is post translationally modified in the mutant embryos. Interestingly, previous experiments using cultured cells have shown that Ric-8B regulates Gαs protein stability by inhibiting its ubiquitination and degradation [8]. Therefore, it is possible that reduced levels of Ric-8B in the mutant embryos leads to increased Gαs ubiquitination, or another type of post-translational modification, and degradation. The presence of strong low-MW bands in the Ric-8B^bgeo/bgeo extracts (asterisk) when compared to Ric-8B^wt/wt and Ric-8B^wt/bgeo extracts could also indicate proteolysis of Gαs.

Interestingly, the IPA analysis showed that the three most significant signaling pathways altered in the Ric-8B model are not from the GPCR signaling pathway, but are related to the initiation and the control of protein synthesis: eIF2, eIF4/P70S6K, and mTOR signaling (Fig 6A, S5 Table, -log(p-value)>1.3), which is in agreement with the GO Biological Process analysis.

## mTOR signaling in Ric-8B mutant embryos

We next assessed the activity of mTOR in E9.5 embryos by the phosphorylation levels of key downstream targets of both complexes. Akt, a prosurvival kinase, is fully activated through the phosphorylation of Thr308 by phosphoinositide-dependent kinase 1 (PDK1) and of Ser473 by mTORC2 [30–33]. We found that while phosphorylation of Akt at Thr308 was not altered in Ric-8B^bgeo/bgeo whole embryo lysates, phosphorylation of Akt (Ser473) catalyzed by mTORC2 was significantly reduced (Fig 6C and 6D). The levels of phosphorylation of ribosomal protein S6, a downstream target for mTORC1, were not significantly altered in Ric-8B^bgeo/bgeo embryos (Fig 6E). These results indicate that the activity of mTORC2, but not of mTORC1, is impaired in Ric-8B^bgeo/bgeo embryos.

Inhibition of mTORC2 function leads to decreased phosphorylation of the FoxO1 and FoxO3a transcription factors, leading to their translocation to the nucleus and subsequent transcription of pro-apoptotic genes [34]. Consistent with this, transcription of the *FoxO3* gene, which is a target for FoxO1/FoxO3 transcription factors [35], is upregulated in the Ric-8B^bgeo/bgeo embryos (log fold change = 0.71; FDR = 0.009; S2 Table). As mentioned above, mTORC2 is also involved in the organization of the cytoskeleton, through the regulation of

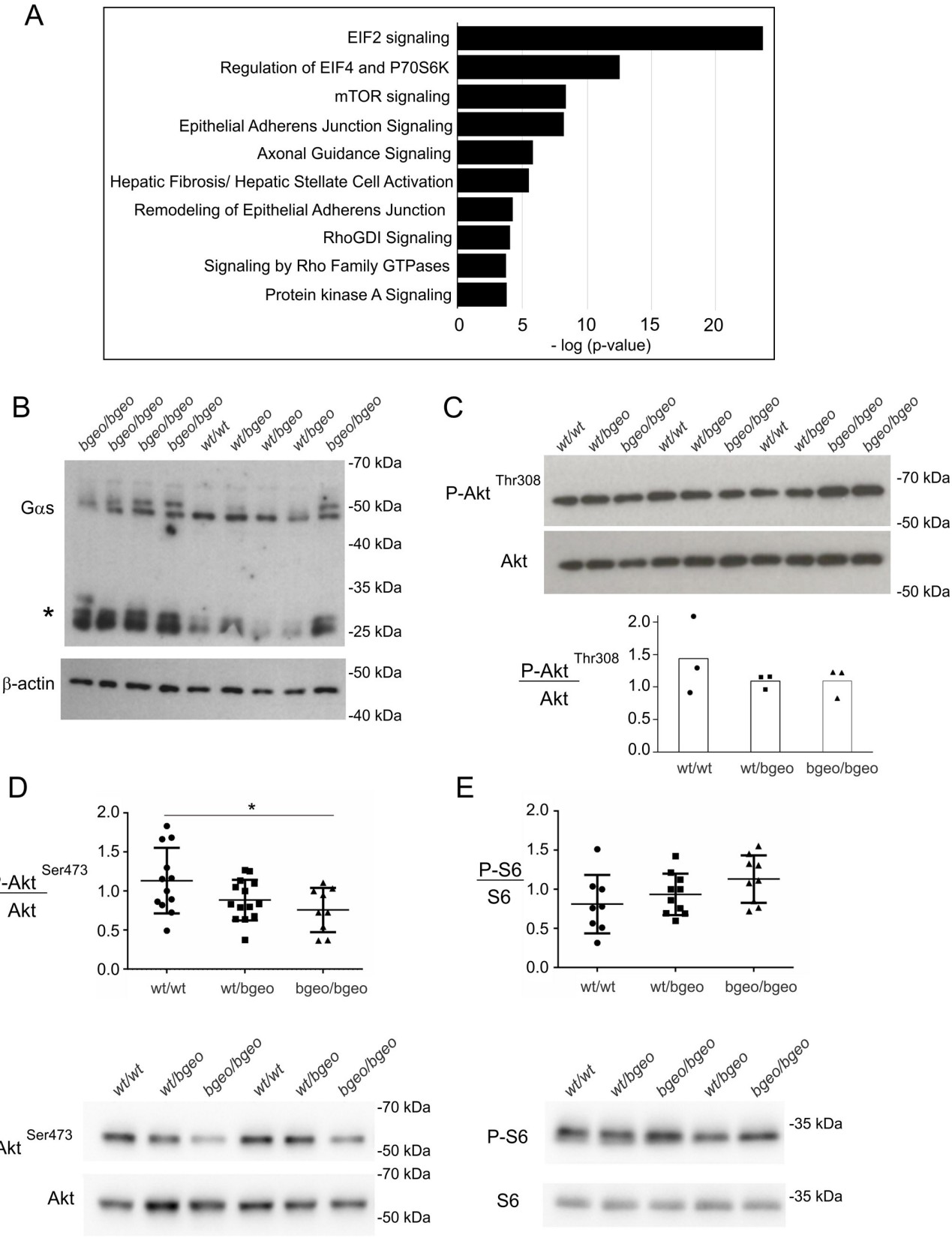

**Fig 6. Phosphorylation of Akt (Ser473) is decreased in Ric-8B$^{bgeo/bgeo}$ embryos.** (A) Cell signaling pathways predicted to be altered in the Ric-8B mutant embryos. Top 10 altered signaling pathways, as predicted by IPA, are shown. The ranking was based on the p values derived from the Fisher's exact test (IPA). The x axis displays the–(log) p value. (B) Western blot of Gαs content in E9.5 embryos from the different Ric-8B genotypes. β-actin was used as loading control. (* possible degradation products of Gαs). (C) Western blot analysis of phospho-Akt (Thr308) in E9.5 embryos compared to pan-Akt. The graph shows the quantification of the ratio of phospho-Akt (Thr308) compared to pan-Akt (D) Semi-quantitative analysis of Akt (Ser473) phosphorylation in E9.5 embryos. The graph shows the quantification of the ratio of phospho-Akt (Ser473) compared to pan-Akt. Values are expressed relative to wild type embryo levels and shown as mean +/- S.E.M. (each dot represents an embryo), *, P<0.05. A representative Western blot is shown below the graph. (E) Semi-quantitative analysis of S6 phosphorylation in E9.5 embryos. The graph shows the quantification of the ratio of phospho-S6 compared to total S6. Values are expressed relative to wild type embryo levels and shown as mean +/- S.E.M. (each dot represents an embryo). A representative Western blot is shown below the graph.

RhoGTPases and PKCα [15]. Notably, signaling pathways involved in actin regulation by Rho GTPases were also identified by the IPA analysis (Fig 6A). Altogether these results indicate that Ric-8B is required for normal mTORC2 activity during mouse embryogenesis.

## mTOR signaling in Ric-8B knockdown cell lines

We next generated conditional shRNA-mediated Ric-8B knockdown HEK293T cell lines and examined mTOR signaling in these cells. shRNAs targeting three different regions of the Ric-8B gene were used (Fig 7A). Western blot analysis showed that expression of the Ric-8B protein was reduced in the three generated cell lines when compared to control cells, with the shRNA17 cell line showing the lowest levels (S6A Fig). Knockdown and control cells were serum starved for 4 hours and then stimulated with 10% FBS, to induce the mTOR signaling pathway. We analyzed the phosphorylation levels of Akt (Ser473) and S6 after FBS stimulation for different time lengths and found that the levels of phospho-Akt (Ser473) are markedly reduced in the shRNA 17 knockdown cells (Fig 7B). Densitometric quantification showed that the levels of S6 phosphorylation are also reduced in the knockdown cells, but not as strikingly as those of Akt (Ser473) (Fig 7B). In addition, the phosphorylation of NDRG1, a substrate of SGK1 and therefore a downstream target of mTORC2 [36], is reduced in the Ric-8B shRNA17 knockdown cells when compared to the control cells (Fig 7C). Altogether, these results show that not only in the mouse embryo, but also in the HEK293T cells, Ric-8B is required mTORC2 activity.

Because previous work showed that Ric-8B stabilizes expression of Gαs/olf protein subunits in different cell types [7,8,11], we asked whether the Gαs protein is stably expressed in our Ric-8B knockdown cells. Experiments using protein extracts prepared from these cells showed that the levels of the Gαs protein are reduced when compared to the levels found in control cells (S6B Fig), raising the possibility that mTORC2 activity depends on Gαs signaling. To address this question, the cells were serum starved and treated with 10% FBS and forskolin to test whether the phosphorylation of Akt (Ser473) could be rescued by increasing the levels of cAMP. As shown in Fig 7D, increasing cAMP did not rescue the low levels of phospho-Akt (Ser473) found in the shRNA17 knockdown cells, and actually, inhibited phosphorylation in the control cells. Inhibition of phosphorylation of Akt (Ser473) by addition of forskolin or cell permeable cAMP analogs in HEK293 cells has already been previously demonstrated [37].

We also asked whether activation or inhibition of Gβγ could rescue mTORC2 activity in the Ric-8B shRNA17 knockdown cells. As before, cells were induced with FBS, but now with the addition of surfen (a putative activator of Gβγ [38]) or gallein (an inhibitor of Gβγ). The addition of surfen did not change the phosphorylation levels in control or knockdown cells, but interestingly, a slight increase in phospho-Akt (Ser473) was observed in the control cells treated with gallein, suggesting that Gβγ may inhibit mTORC2 activity in wild type cells (Fig 7E). On the other hand, gallein did not rescue mTORC2 activity in the Ric-8B shRNA17 cells (Fig 7E), indicating that Gβγ may not be involved in mTORC2 regulation in these cells.

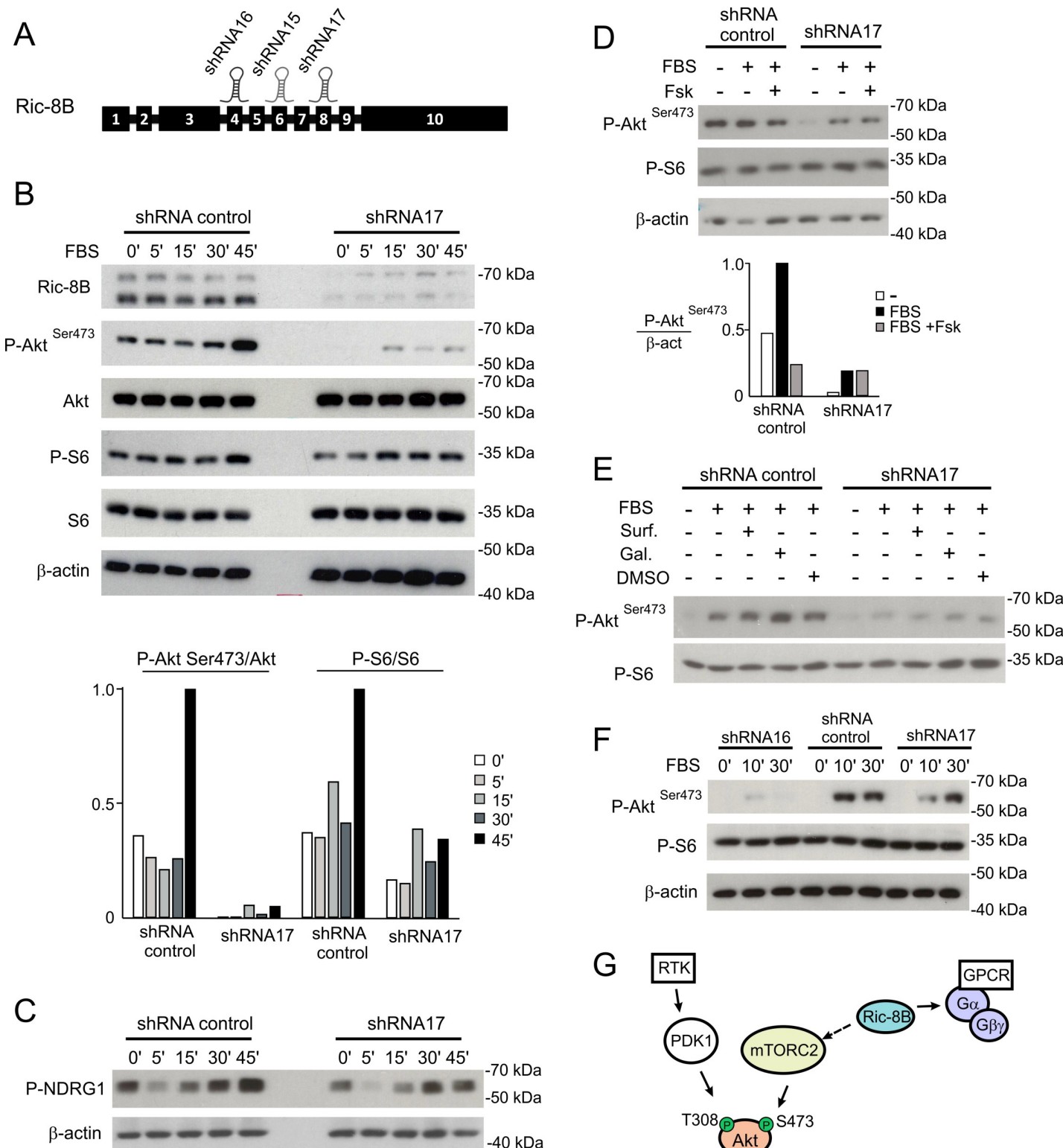

**Fig 7. Ric-8B is required for mTORC2 activation in cultured cells.** (A) Schematic representation of the Ric-8B gene structure showing the regions targeted by the shRNAs used to knockdown endogenous expression of Ric-8B in cultured cells. (B) HEK293T cell lines containing a control shRNA (shRNA for luciferase) or the Ric-8B shRNA17 were starved for 4 hours and stimulated with FBS for 5, 15, 30 or 45 minutes. Total protein lysates were analyzed in Western blot experiments for the expression of the indicated proteins. The graph shows the quantification of the Western blots by densitometry of the ratio of phospho-Akt (Ser473) compared to total Akt and of phospho-S6 compared to total S6. (C) The same protein extracts as in (B) were analyzed in Western blot experiments for pNDRG1, a downstream target of

mTORC2. (D) HEK293T cell lines containing the control shRNA or the Ric-8B shRNA17 were starved for 4 hours and stimulated with FBS or FBS plus 20μM forskolin for 30 minutes. Total protein lysates were analyzed in Western blot experiments for the expression of the indicated proteins. The graph shows the quantification by densitometry of the ratio of phospho-Akt (Ser473) compared to β-actin. (E) HEK293T cell lines containing the control shRNA or the Ric-8B shRNA17 were starved for 4 hours and stimulated with FBS or FBS plus 10μM surfen or 20 μM gallein for 30 minutes. Total protein lysates were analyzed in Western blot experiments for the expression of the indicated proteins. (F) HepG2 cell lines containing the control shRNA, Ric-8B shRNA16 or Ric-8B shRNA17 were starved for 4 hours and stimulated with FBS for 10 or 30 minutes. Protein lysates were analyzed for the expression of the indicated proteins as described above. (G) Schematic representation of the role played by Ric-8B in mTORC2 regulation. Growth factors activate PI3K at the cell membrane leading to the production of PIP3, recruitment of Akt and PDK1 and phosphorylation of Akt (Thr308) by PDK1. Depletion of Ric-8B impairs function and/or stability of Gα and Gβγ subunits and reduces phosphorylation of Akt (Ser473) by mTORC2. Increase in cAMP does not rescue mTORC2 activity, indicating that the effect is not due to deficient Gαs function. Whether Gβγ subunits are required for mTORC2 still needs to be determined. Ric-8B may also regulate mTORC2 independently of G proteins (dashed arrow). Phosphorylated substrates analyzed in our experiments are indicated in green.

Another possibility is that the knockdown cells lack functional Gβγ subunits, due to the reduced expression of Gαs (S6B Fig). Additional experiments are required to clarify whether Gβγ is involved in regulation of mTORC2, but our results so far indicate that the reduced activity of mTORC2 observed in the Ric-8B shRNA17 knockdown cell line is not a consequence of reduced cAMP signaling.

Finally, we asked whether the role played by Ric-8B in mTORC2 activity can be extended to other cell types. We knocked down Ric-8B in the HepG2 cell line (a human hepatoma cell line) and analyzed Akt (Ser473) phosphorylation in these cells. We found that in HepG2 knockdown cells, mTORC2 activity is also reduced (Fig 7F), indicating therefore, that the role played by Ric-8B in mTORC2 function is not restricted to HEK293T cells.

## Discussion

In this study, we show that the Ric-8B hypomorph embryos fail to close the neural tube at the cephalic region. Our results show that increased apoptosis occurs in the region of the neural tube defect in Ric-8B mutant mice. Expansion of the cranial mesenchyme is required for the elevation of the cranial neural folds, and excessive apoptosis in the mesenchyme would preclude this expansion. Therefore, excessive apoptosis must cause or contribute to the cranial neural tube defect observed in the Ric-8B mutant mice. The finding that MEF cultures generated from mutant Ric-8B embryos also show increased frequency of apoptotic cells supports a role for Ric-8B in processes related to apoptosis or cell survival. We found that mTORC2 activity, but not mTORC1 activity, is reduced in the Ric-8B mutant embryos when compared to wild type embryos. Since mTORC2 phosphorylation of Akt (Ser473) promotes cell survival versus apoptosis [39], decreased activity of mTORC2 could lead to increased apoptosis in the Ric-8B mutant embryo. However, further experiments are required to determine whether apoptosis is a direct consequence of mTORC2 dysregulation in this case.

Open cephalic neural tube phenotypes are also seen in embryos that are mutant for a series of different genes, indicating that multiple processes are required for successful closure of the neural tube [27,28]. Notably, embryos deficient for the G protein Gβ1 subunit also exhibit similar defects [40], supporting the involvement of G protein mechanisms in these processes. Mutations that lead to the upregulation of the Shh pathway also may result in open cranial tube phenotypes [41]. For example, complete loss of the ciliary Gpr161 leads to increased Shh signaling and lethality by E10.5 with open forebrain and midbrain regions [42]. Loss of Gαs leads to activation of Shh signaling and embryos die at E9.5 with open neural tube and cardiac defects [43]. We found however, that Shh signaling is not severely altered in the Ric-8B homozygous mutants. Still, since these Ric-8B mutants are not null, we cannot exclude the possibility that the residual function of Ric-8B may attenuate the actual Ric-8B null phenotype. Accordingly, knockout of the Ric-8B gene results in embryonic lethality at earlier stages (between E4

and E8.5), indicating that in this case the mice present earlier phenotypes that result in earlier death [7].

Mutant mice have already been generated to study the role played by the mTOR complexes *in vivo*. Interestingly, embryos that are ablated for *Rictor*, a specific component present in mTORC2 that is required for its function, are smaller than their littermates [32,33], a phenotype that is also shown by Ric-8B mutant embryos (Fig 4J). Even though *Rictor* null embryos do not show open cranial neural tubes, mice knocked out for *mLST8* fail to develop the cephalic region [32]. mLST8 contains seven WD-40 repeats that are common to Gβ subunits, and is therefore also named GβL (G protein β subunit like protein) [44]. mLST8 is present in both mTORC1 and mTORC2 complexes, but required only for mTORC2 function during development. *Rictor* and *mLST8* deficient embryos are phenotypically similar, they die around E10.5 and show reduced phosphorylation of Akt (Ser473) [32,33]. In addition, in these mutant embryos, phosphorylation of Akt (Thr308) is not altered when compared to the wild type embryos, as also observed for the Ric-8B mutant embryo [32,33].

Despite the similarities, the differences in the phenotypes of the *Rictor* and *Ric-B* mutant embryos are likely to result from additional pathways that are altered in these two models. Nevertheless, many signaling pathways altered in Ric-8B[bgeo/bgeo] embryos can be associated with mTORC2 function. Genes coding for eIFs and ribosome proteins are upregulated in the Ric-8B[bgeo/bgeo] embryo (S1 Table and S2 Table), and account for the major part of the differentially expressed genes associated to EIF2 signaling and regulation of EIF4 and P70S6K pathways (Fig 6A, S5 Table). Although mTORC1 has been known as the bona fide regulator of protein synthesis, emerging evidence suggest that this process can also be controlled by mTORC2 [45]. Accordingly, ribosome biosynthesis and assembly are enriched in our GO Biological Process analysis (S4 Table, adjusted p-value <0.05). Epithelial adherens junction signaling, remodeling of epithelial adherens junctions, and signaling by Rho GTPases (Fig 6A) can also be related to mTORC2 signaling in the Ric-8B model. Rho small GTPases RhoA, Rac1 and Cdc42 are known to regulate junctions in cell–cell adhesion, and were shown to mediate mTORC2 control over the reorganization of actin cytoskeleton in different cell lines [17,46]. Curiously, the hepatic fibrosis/hepatic stellate cell activation is also one of the top altered IPA signaling pathways (Fig 6A). This process is a wound healing response to injury characterized by deposition of extracellular matrix mainly by hepatic stellate cells, and it has been shown to be activated by the PI3K/AKT/ mTOR pathway [47]. The liver phenotype of the Ric-8B[bgeo/bgeo] embryos was however not analyzed.

Altogether these findings suggest that Ric-8B mutant embryonic phenotypes may be, at least in part, consequence of impaired mTORC2 signaling. It is important to note that we used whole-body embryos to generate our transcriptome data, and therefore we cannot exclude from our gene expression analysis some possible secondary effects of Ric-8B depletion (e.g. cells that do not express Ric-8B but are affected by Ric-8B expressing cells, or more complex phenotypes derived from the lack of Ric-8B in earlier stages). Nevertheless, the role of Ric-8B in mTORC2 activity is strongly supported by our experiments using two different cell lines, where reduction of phospho-Akt (Ser473) levels were consistently observed after Ric-8B knockdown.

Upstream regulators of mTORC2 are still largely unknown [15,18,48,49]. Recent studies have found that adrenergic signaling may induce mTORC2 activation [50–52]. One of these studies showed that Akt (Ser473) phosphorylation can be induced in brown adipocytes upon β-adrenergic stimulation in a PI3K-dependent and mTORC1-independent fashion [50], indicating that GPCR signaling through Gs can activate mTORC2 at least in some cell types. Even though we found that Ric-8B knockdown cells express lower levels of Gαs, addition of a cell permeable cAMP analog to the cells did not rescue the phospho-Akt (Ser473) levels, indicating

that the mechanism by which the Ric-8B knockdown impairs mTORC2 activity does not depend on cAMP signaling in HEK293T cells.

Growth factors can activate mTORC2 through PI3K by mechanisms that are still not completely understood [20,21]. GPCR-promoted activation of PI3K is thought to be mediated by Gβγ subunits [53–55], and a study also showed that gallein, a pharmacological inhibitor of Gβγ, can attenuate the phosphorylation of Akt (Ser473) induced by the chemokine receptor CXCR4 [56]. However, inducing or inhibiting Gβγ signaling in the Ric-8B knockdown cells did not restore the phospho-Akt (Ser473) levels, raising the possibility that Ric-8B may have additional targets that are unrelated to G proteins and are yet to be identified (Fig 7G).

In conclusion, our results show that Ric-8B is required for proper growth and nervous system formation during mouse embryogenesis. They also uncover a novel link between Ric-8B and mTORC2 signaling. Extensive studies of the Ric-8B mutants should contribute to unravel the mechanisms that regulate mTORC2 activity, which are still little understood.

## Materials and methods

### Animal procedures

All procedures undertaken in this study were approved by the University of São Paulo Chemistry Institute's Animal Care and Use Committee, under the protocol #19/2013 and #60/2017.

### Gene trap mice

The Baygenomics ES cell lines RRH188 and RRA103 derived from the 129P2Ola/HSD strain were used. The insertion of the gene trap vectors (pGT01xf and pGT11xf, respectively) in the Ric-8B gene was confirmed by RT-PCR, using the vector primer β-geo R together with primer RRH188F in exon 3 or primer RRA103F in exon 7, to detect the corresponding chimeric mRNAs. The cells were injected into C57BL/6 strain blastocysts at the Mutant Mouse Regional Resource Center (MMRRC) at the University of California, Davis (http://www.mmrrc.org) to produce chimeric mice. Chimeric male mice showing high percent chimerism were crossed to C57BL/6 females and the resulting agouti offspring was genotyped by PCR on ear genomic DNA using primers that recognize the gene trap vector (vector F2, and β–geo R). Primer sequences are shown in S3 Fig. The precise site of vector insertion in the third intron of Ric-8B was determined as described in S2 Fig. Heterozygous mice were intercrossed and the resulting mice were genotyped by a multiplex PCR using three primers: the forward primer 188intronF2 in intron 3 and two reverse primers, the VectorR2 primer in the gene trap vector for the Ric-8B KO allele (312 bp PCR product), and the 188intronR2 primer in intron 3 for the WT allele (582 bp PCR product). PCR reactions started with a denaturation step of 95°C for 3 minutes, followed by 35 cycles of 95°C for 45 s, 68°C for 45 s and 72°C for 1 minute.

### X-gal staining of tissues and embryos

X-gal staining was performed as described in [57]. Briefly, for whole mounts, tissues were dissected and incubated on ice for 30 minutes with 100 mM phosphate buffer (pH 7.4), 4% PFA, 2 mM $MgSO_4$, 5 mM EGTA, washed once with 100 mM phosphate buffer (pH7.4), 2 mM $MgCl_2$, 5mM EGTA at room temperature and further incubated for 30 minutes with the same buffer. Tissues were then washed twice for 5 minutes at room temperature with 100 mM phosphate buffer (pH 7.4), 2 mM $MgCl_2$, 0.01% sodium deoxycholate, 0.02% NP-40 and incubated for 3 hours (olfactory epithelium), 9 hours (striatum) or overnight (embryos) in the dark at 37°C with 100 mM phosphate buffer (pH 7.4), 2 mM $MgCl_2$, 0.01% sodium deoxycholate, 0.02% NP-40, 5 mM potassium ferricyanide, 5 mM potassium ferrocyanide, 1mg/ml X-gal.

For sections, whole mount embryos stained with X-gal were frozen to -20˚C in OCT (Sakura TissueTek) and immediately sectioned using a cryostat. It is important to note that we also detected endogenous β-galactosidase activity in the olfactory epithelium of wild type mice, as previously described [58], however, it was observed only after one overnight incubation with X-gal, while staining of the olfactory epithelium from heterozygous RRH188 mice is visible after 3 hours.

### *In situ* hybridization

Dissected embryos were fixed in 4% PFA for 16 hours at 4˚C. Cryopreservation was performed in 30% sucrose, 50% OCT for 2 hours at 4˚C. Then, whole mount embryos were frozen in OCT and sectioned using a cryostat. For *in situ* hybridization experiments, sections were fixed in 4% PFA for 10 minutes, washed twice in PBS for 5 minutes and digested with Proteinase K 10 μg/ml for 8 minutes. Then, sections were fixed in 4% PFA for 15 minutes and washed twice in PBS for 5 minutes. The following steps were performed as described in [2].

### Western blot

Whole E9.5 embryos or cultured cells were lysed with cold RIPA buffer (100 mM Tris pH 7,4; 0,25% sodium deoxycholate; 150 mM NaCl, 1 mM EDTA and 1% NP40 v/v containing 1 X phosphatase inhibitor phoSTOP (Sigma #4906845001) and 1 X protease inhibitor cocktail (Sigma # P2714) and 10% SDS-PAGE was used to fractionate proteins. Western blotting was performed by using anti-Gli3 (Proteintech #19949-1-AP), anti-Ric-8B (Atlas #HPA042746), anti-phospho-S6 (Cell Signaling #5364), anti S6 (Cell Signaling #2217), anti-phospho-Akt (Ser473) (Cell Signaling #4060), anti-phospho-Akt (Thr308) (Cell Signaling#2965), anti-Akt 1 (Millipore #06–885), anti-Akt (pan) (Cell Signaling #2920), anti-Gαs (Santa Cruz #55545), anti-α-tubulin (Sigma #T5168) or anti-β-actin (Santa Cruz #47778) antibodies. Immunoreactivity was detected by incubating the membranes with specific HRP-conjugated secondary antibodies (Cell Signaling), and visualized using a chemiluminescence system (Amersham ECL Prime, GE Healthcare #RPN2133). For the semi-quantitative analysis of phospho-AKT (Ser473) and phospho-S6 in the embryonic extracts, the blots were scanned using a gel image capture system to quantify differences via densitometry (Alliance 2.7 system, Alliance 1D capture software, and UVIBand 12.14 analysis software; UVITEC, Cambridge, UK). Values are normalized with β-actin. Data are expressed as mean ± standard error of the mean (SEM). Statistical analysis was performed by one-way ANOVA followed by Tukey's post hoc test.

### RT-PCR

RNA was purified from mouse embryos and cDNA synthesis was performed as previously described [2]. The pair of primers Ric-8BFexon3 and Ric-8BRexon5, or RicRTF and RicRTR (see S3 Fig) were used to amplify different regions of the Ric-8B cDNA. The PCR reaction was carried out for 28 cycles, and the PCR products were analyzed in 1% agarose gels.

### Real-time PCR

Ric-8B gene expression was quantified by real-time PCR using the ABI (USA) 7300 Real-Time PCR system, and cDNAs were prepared from E10.5 embryos as previously described [2,59]. Primer sequences were Ric-8BF *forward* 5'-AGCTGGTTCGTCTCATGACAC-3' and Ric-8BR *reverse* 5'-CAGCGTTCCCATAGCCAGTG-3'. All reactions were performed by using a standard real time PCR protocol (1 cycle of 95˚C for 10 min, 40 cycles of 95˚C for 15 s, and 60˚C for 1 min). Data was normalized by using β-actin as reference. Relative gene expression

between different embryos was then calculated as $2^{-\Delta\Delta Ct}$, using the sample of the wild type embryo as calibrator, according to Livak and Schmittgen [60]. Each reaction was performed in triplicate and the standard deviation was inferior to 0.3.

## Scanning electron microscopy

Embryos were dissected in cold PBS and fixed overnight at 4˚C in PBS containing 4% PFA, 2.5% glutaraldehyde, 1% tannic acid and 5 mM $CaCl_2$. The embryos were dehydrated through a graded ethanol series and critical point-dried (Balzers CPD 050). After gold-sputtering (20 nm-gold layer) (BalTec CSC 020), the embryos were observed in a Jeol 5310 operating at 15 kV. Digital images were acquired at the resolution of 1024×770 pixels using SEMafore software.

## Immunostaining

Embryos were fixed overnight at 4˚C in 4% PFA, followed by cryoprotection in 30% sucrose, and embedding in OCT (Sakura TissueTek). OCT blocks were sectioned at 14–16 μm using a cryostat. Sections were blocked in 10% FBS, 0.2% Triton-X for 1 hour, followed by incubation with anti-cleaved caspase-3 (Cell Signaling #9661, 1:250) diluted in 1% FBS, 0.2% Triton-X. After incubation, sections were washed with PBS and incubated with Alexa488-conjugated donkey anti-rabbit IgG (Life Technologies #A-21206, 1:300) and Rhodamine Phalloidin diluted in 1% FBS, 0.2% Triton-X. Nuclei were stained with DAPI (SIGMA, 0.5 μg/ml).

## MEFs

MEFs were isolated from E9.5 embryos of each genotype and cultured in DMEM with 10% FBS. Cells were pulsed for 1 hour with 10μM bromodeoxyuridine (Brdu) and fixed in 4% PFA for 20 minutes at room temperature. After fixation, cells were double stained with anti-BrdU (Santa Cruz sc-32323, 1:400) and anti-cleaved caspase-3 primary antibodies (Cell Signaling #9661, 1:250). Nuclei were stained with DAPI (SIGMA, 0.5 μg/ml). Photographs were taken using a Nikon TE300 microscope.

## RNA sequencing (RNA-Seq) and signaling pathway analysis

Ribosomal RNA was depleted from total RNA prepared from E10.5 mouse embryos using the RiboMinus Eukaryote Kit for RNA-Seq (Invitrogen). RNA-Seq libraries were prepared using SOLiD Total RNA-Seq Kit, according to the manufacturers' recommendations, and were sequenced on the SOLiD sequencing platform (Life technologies, Carlsbad, CA). Sequences were then aligned to the mouse genome (version GRCm38/mm10) using Tophat [61] with default parameters and gene annotations provided by Ensembl (version 71, [62]). Alignments were filtered with SAMtools [63]. The uniquely mapped reads with minimum mapping quality 20 were used to calculate gene expression, which was generated using Cufflinks [64]. Pathway analysis was performed using Ingenuity Pathway Analysis (IPA, QIAGEN, Redwood City, www.qiagen.com/ingenuity). The overrepresented pathways, shown in Fig 6A, met the IPA cut-off threshold for significance (p-value < 0.05). Analysis was run by using all default settings for the selection of dataset, no fold-change cut-off, FDR <0.05 and P-value < 0.05.

## Knockdown experiments

To knockdown Ric-8B expression in cultured cells we used the SMARTvector Inducible Human RIC8B shRNA (V3SH11252-227737126 GGATGTTTCGATGGGCTCG; V3SH11252-229071976 TAAACAATGACGAAGGACA; V3SH11252-229376731 CTGAG-TACCAATTATCTCC (Dharmacon). For the experiments shown in Fig 7 cells were plated in

Dulbecco's Modified Eagle's Medium (DMEM) supplemented with 10% of fetal bovine serum (FBS) and doxycycline (1–4 μg/mL). Three days later, medium was removed and cells were washed twice with PBS 1x. Cells were starved in DMEM without FBS for 4 hours and then stimulated with DMEM 10% FBS (or DMEM 10% FBS plus 20 μM forskolin, 20 μM gallein or 10 μM surfen) for different time lengths.

## Supporting information

**S1 Fig. Expression of Ric-8B as revealed by β-galactosidase activity in different mouse tissues.** The table shows the results obtained by whole-mount X-gal staining of adult Ric-8B$^{wt/bgeo}$ mouse tissues. (+) blue staining; (-) no staining was detected. Some of the analyzed tissues (prostate, intestine, kidney, testis and seminal vesicle) showed endogenous β-galactosidase activity.
(TIF)

**S2 Fig. Identification of the precise site of vector insertion in the Ric-8B gene.** The locations of the primers used for the identification of the site of insertion of the gene trap vector are indicated. Different pairs of primers were used in PCR reactions with genomic DNA prepared from heterozygous mice as indicated. PCR products were only obtained for the 188intronF1/ Vector R and 188intronF2/ Vector R pairs of primers, indicating that the vector is inserted ~150 bp downstream to the region matched by the primer 188intronF2.
(TIF)

**S3 Fig. Primer sequences.** List of primer sequences used for genotyping of the embryos and RT-PCR experiments.
(PDF)

**S4 Fig. Shh signaling in Ric-8B $^{bgeo/bgeo}$ embryos.** (A) Transverse sections cut through the neural tubes of X-gal stained Ric-8B$^{wt/bgeo}$ embryos at different developmental stages shows that β-galactosidase activity is restricted to the floor plate. (B) The expression of the floor plate markers Shh and FoxA2, in addition to Ptch1, a direct target of the Shh signaling pathway, was indistinguishable between the neural tubes from E9.5 Ric-8B$^{wt/wt}$ and Ric-8B$^{bgeo/bgeo}$ embryos, indicating that the most ventral neural types are normally specified in Ric-8B$^{bgeo/bgeo}$ embryos. Transverse section cut through the neural tubes were hybridized with antisense probes specific for Shh, FoxA2 and Ptch1. (C) The Gli3 protein is one of the major transcription factors that mediate the transcriptional effects of Shh signaling. In the absence of Shh signaling, Gli3 is proteolytically processed to produce a form that acts as a transcriptional repressor [65]. Western blotting with antibody against Gli3 was used to analyze total protein extracts prepared from E9.5 whole embryos. The amounts of both the activator (Gli3 FL, 230 kDa) and repressor (Gli3 R, 83 kDa) forms of Gli3 in Ric-8B$^{bgeo/bgeo}$ embryos are not different from the ones shown by wild type or heterozygous embryos. Quantification of relative amounts of Gli3 FL and Gli3 R normalized to respective α-tubulin levels is shown at the bottom of the blot. Gli3 FL (Gli3 full length); Gli3 R (Gli3 repressor).
(TIF)

**S5 Fig. Signaling pathways involving G proteins altered in the Ric-8B mutant embryos.** (A) Gα subunits and Ric-8B gene expression in the embryo. Sections cut through the neural tube of an E10.5 wild type embryo were hybridized with digoxigenin-labeled antisense RNA probes for Ric-8B, Gαolf and Gαs, as indicated. (B) Signaling pathways involving G proteins that are altered in the Ric-8B mutant embryos, as identified by IPA, are shown.
(TIF)

**S6 Fig. Ric-8B and Gαs protein expression in the Ric-8B knockdown cell lines.** (A) HEK293T cell lines transfected with control shRNAs (SCR, Scrambled or LUC, luciferase) or with the different shRNAs targeting Ric-8B (shRNA15, shRNA16 and shRNA17) were treated with doxycycline. Total lysates prepared from these cells were analyzed in Western blot experiments for the expression of Ric-8B. Doxycycline (Dox) concentrations used to induce shRNA expression are indicated. (B) Total lysates prepared from HEK293T cell lines expressing a control shRNA, Ric-8B shRNA16 or Ric-8B shRNA17 were analyzed in Western blot experiments for the expression of Gαs. The amount of Gαs/β-actin proteins were quantitated by densitometric analysis and are shown relative to the amount found in the control cells. (C) HepG2 knockdown cells were analyzed for the expression of Ric-8B as described in (A).
(TIF)

**S1 Table. List of all significant downregulated genes in Ric-8B$^{bgeo/bgeo}$ embryos.**
(XLS)

**S2 Table. List of all significant upregulated genes in Ric-8B$^{bgeo/bgeo}$ embryos.**
(XLS)

**S3 Table. Gene ontology analysis of the differentially expressed genes using MGI Mammalian Phenotype.**
(XLSX)

**S4 Table. Gene ontology analysis (Biological Process).**
(XLSX)

**S5 Table. Ingenuity Pathway Analysis (IPA).**
(XLSX)

## Acknowledgments

We are very grateful to Chao Yun Irene Yan, Deborah Schechtman, José Xavier Neto and Hiroaki Matsunami for helpful comments and suggestions. We also thank Silvania Neves and Renata Spaluto for assistance with animal care and Erica Bandeira for technical assistance and Lucas Ferreira da Silva for valuable recommendations for data analysis.

## Author Contributions

**Conceptualization:** Maíra H. Nagai, Victor P. S. Xavier, Luciana M. Gutiyama, Jose G. Abreu, William T. Festuccia, Bettina Malnic.

**Data curation:** Maíra H. Nagai, Elisa R. Donnard, Pedro A. F. Galante.

**Formal analysis:** Maíra H. Nagai, Victor P. S. Xavier, Luciana M. Gutiyama, Elisa R. Donnard, Pedro A. F. Galante.

**Funding acquisition:** Bettina Malnic.

**Investigation:** Maíra H. Nagai, Victor P. S. Xavier, Luciana M. Gutiyama, Cleiton F. Machado, Alice H. Reis, William T. Festuccia.

**Methodology:** Maíra H. Nagai, Victor P. S. Xavier, Luciana M. Gutiyama, Bettina Malnic.

**Project administration:** Bettina Malnic.

**Resources:** Bettina Malnic.

**Supervision:** Jose G. Abreu, William T. Festuccia, Bettina Malnic.

**Validation:** Maíra H. Nagai, Victor P. S. Xavier, Bettina Malnic.

**Visualization:** Maíra H. Nagai, Victor P. S. Xavier, Luciana M. Gutiyama, Alice H. Reis, Bettina Malnic.

**Writing – original draft:** Maíra H. Nagai, Bettina Malnic.

**Writing – review & editing:** Maíra H. Nagai, Victor P. S. Xavier, Luciana M. Gutiyama, Alice H. Reis, Pedro A. F. Galante, Jose G. Abreu, William T. Festuccia, Bettina Malnic.

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
