## [Decision Letter · Decision Letter 0]

7 Aug 2019

Dear Dr. Bettina Malnic,

Thank you very much for submitting your Research Article entitled 'Depletion of Ric-8B leads to reduced mTORC2 activity' to PLOS Genetics. Your manuscript was fully evaluated at the editorial level and by two independent peer reviewers that are experts in the field. The reviewers appreciated the attention to an important problem, but raised some substantial concerns about the current manuscript. Based on the reviews, we will not be able to accept this version of the manuscript, but we would be willing to review again a much-revised version. We cannot, of course, promise publication at that time.

Should you decide to revise the manuscript for further consideration here, your revisions should address the specific points made by each reviewer. We will also require a detailed list of your responses to each of the review comments and a description of the changes you have made in the manuscript.

If you decide to revise the manuscript for further consideration at PLOS Genetics, please aim to resubmit within the next 60 days, unless it will take extra time to address the concerns of the reviewers, in which case we would appreciate an expected resubmission date by email to plosgenetics@plos.org.

[LINK]

We are sorry that we cannot be more positive about your manuscript at this stage. Please do not hesitate to contact us if you have any concerns or questions.

Yours sincerely,

J. Silvio Gutkind

Guest Editor

PLOS Genetics

Gregory Barsh

Editor-in-Chief

PLOS Genetics

Reviewer's Responses to Questions

**Comments to the Authors:**

Reviewer #1: Review: PLOS Genetics

Depletion of Ric-8B leads to reduced mTORC2 activity

The manuscript authored by M. H. Nagai et al. describes the phenotype of Ric-8B hypomorph embryos. The authors found that the embryos are smaller than their wild type littermates, that they fail to close their neural tube in the cephalic region, and that they die during mid-embryogenesis. Not surprisingly transcriptome analysis revealed that signaling pathways involving GPCRs and G proteins are dysregulated in the Ric-8B mutant embryos. Interestingly, this analysis also revealed an unexpected impairment of the mTOR signaling pathway. Supporting this observation, the phosphorylation of Akt at S473 was downregulated in the Ric-8B mutant embryos suggesting a decreased activity of mTORC2 and knockdown of Ric-8B in HEK293T cells led to a reduction of the basal level of S473 Akt and a poor induction following serum starvation and re-exposure.

Comments

The authors description of the phenotype of the Ric-8B hypomorph embryos is the focus of the first 5 figures. Evidently enough Ric-8B is expressed in these embryos to allow them to progress slightly further in development than had been previously noted following complete loss of Ric-8B. While the absence of Ric-8A on mouse development has been reported previously the phenotype of the Ric-8B-/- embryos was not previously examined in any detail. The authors convincingly show that Ric-8B is required for neural tube closer.

Figure 6 summarizes the transcriptome analysis and shows a global reduction in pS473 Akt in the e9.5 embryos via immunoblotting. The authors should show molecular weight markers, quantitate their blots, and immunoblot the lysates to document G-alpha s expression and examine pS308 Akt.

Figure 7 shows that the Ric-8B KD in HEK293T results in a loss of Gs and a reduction in pS473 Akt. Again, the authors should quantitate their results. At a minimum they should explore in more detail the mechanism by which the Ric-8B KD impairs mTORC2 activity. Does it depend on Gs signaling, is cAMP involved, or PKA?

Reviewer #2: This study by Nagai et al investigates the in vivo function of Ric-8B, which encodes a GEF for Galpha(s) and Galpha(olf). To do this, Ric-8B gene trap cells were obtained and mice homozygous for the gene trap allele were generated. The authors describe an interesting phenotype related to neural tube defects and cell survival. Signaling experiments suggest a defect in AKT phosphorylation that could be related to a survival defect, and a defect in Galpha(s) levels, though the two are not connected. Evidence for Ric-8B function in AKT signaling (through mTORC2) is also provided in HEK293T cells and MEFs. However, the study falls short of explaining what the connection is. Additional comments below:

1. Other than the embryos dying around the same time as mTORC2-deficient embryos, they don’t seem that similar. The authors need to better explain this discrepency.

2. I am not sure what the statement means (line 161-163) that “the residual levels present in the homozygous embryos is likely to result from the gene knockout technology used.” Could the authors elaborate?

3. Could the authors provide more discussion about which genes are affected in the Ric-8B model? And how these gene expression differences are connected to a defect in mTORC2 or AKT activity? Is there any caveat of doing transcriptome analysis on whole embryos rather than on the cells with high Ric-8B expression?

4. It does look like there is some effect on AKT-S473 in embryos, but it does not look that strong. Could the authors look specifically in the cells where Ric-8B is most highly expressed?

5. The results in HEK293T cells is interesting as it could suggest a broader role for Ric-8B, but the early time points following FBS treatment do not show dynamic changes in AKT phosphorylation. Are these the right cells to use for these experiments? This could be addressed by examining a few cell lines in which AKT shows sensitivity to serum deprivation and FBS stimulation. Also, do these cells have survival defects? Other defects indicative of a loss in mTORC2 signaling (e.g. PKC levels, SGK activity)?

6. How mTORC2 activity might be selectively decreased by Ric-8B loss is not explained. There is an interesting correlative decrease in Galpha(s) protein levels. Can defective AKT phosphorylation and survival be rescued through this pathway?

**Have all data underlying the figures and results presented in the manuscript been provided?**

Reviewer #1: Yes

Reviewer #2: Yes

PLOS authors have the option to publish the peer review history of their article (what does this mean?). If published, this will include your full peer review and any attached files.

Reviewer #1: No

Reviewer #2: No

---

## [Decision Letter · Decision Letter 1]

24 Feb 2020

Dear Dr Malnic,

We are pleased to inform you that your manuscript entitled "Depletion of Ric-8B leads to reduced mTORC2 activity" has been editorially accepted for publication in PLOS Genetics. Congratulations!

Yours sincerely,

J. Silvio Gutkind

Guest Editor

PLOS Genetics

Gregory Barsh

Editor-in-Chief

PLOS Genetics

Comments from the reviewers (if applicable):

Reviewer's Responses to Questions

**Comments to the Authors:**

Reviewer #1: The authors have satisfied the majority of my concerns.

**Have all data underlying the figures and results presented in the manuscript been provided?**

Reviewer #1: Yes

PLOS authors have the option to publish the peer review history of their article (what does this mean?). If published, this will include your full peer review and any attached files.

Reviewer #1: Yes: John Kehrl

**Data Deposition**

http://datadryad.org/submit?journalID=pgenetics&manu=PGENETICS-D-19-00958R1

**Press Queries**

---

## [Editor Report · Acceptance letter]

30 Apr 2020

PGENETICS-D-19-00958R1 

Depletion of Ric-8B leads to reduced mTORC2 activity 

Dear Dr Malnic, 

We are pleased to inform you that your manuscript entitled "Depletion of Ric-8B leads to reduced mTORC2 activity" has been formally accepted for publication in PLOS Genetics! Your manuscript is now with our production department and you will be notified of the publication date in due course.

With kind regards,

Kaitlin Butler

PLOS Genetics

On behalf of:
